

# Devil in the details: how can we avoid potential pitfalls of CATS regression when our data do not follow a Poisson distribution?

Zoltán Botta-Dukát

Centre for Ecological Research, Vácrátót, Hungary

## ABSTRACT

**Background.** Community assembly by trait selection (CATS) allows for the detection of environmental filtering and estimation of the relative role of local and regional (meta-community-level) effects on community composition from trait and abundance data without using environmental data. It has been shown that Poisson regression of abundances against trait data results in the same parameter estimates. Abundance data do not necessarily follow a Poisson distribution, and in these cases, other generalized linear models should be fitted to obtain unbiased parameter estimates.

**Aims.** This paper discusses how the original algorithm for calculating the relative role of local and regional effects has to be modified if Poisson model is not appropriate.

**Results.** It can be shown that the use of the logarithm of regional relative abundances as an offset is appropriate only if a log-link function is applied. Otherwise, the link function should be applied to the product of local total abundance and regional relative abundances. Since this product may be outside the domain of the link function, the use of log-link is recommended, even if it is not the canonical link. An algorithm is also suggested for calculating the offset when data are zero-inflated. The relative role of local and regional effects is measured by Kullback-Leibler $R^2$. The formula for this measure presented by *Shipley (2014)* is valid only if the abundances follow a Poisson distribution. Otherwise, slightly different formulas have to be applied. Beyond theoretical considerations, the proposed refinements are illustrated by numerical examples. CATS regression could be a useful tool for community ecologists, but it has to be slightly modified when abundance data do not follow a Poisson distribution. This paper gives detailed instructions on the necessary refinement.

## INTRODUCTION

The community assembly by trait selection (CATS) method developed by Shipley et al. allows for the detection of environmental filtering of traits without using data on acting environmental variables (*Shipley, Vile & Garnier, 2006*; *Shipley, 2010*). Moreover, its extension (*Shipley, 2014*) can estimate the relative role of environmental filtering, meta-community effects (mass effect or dispersal limitation), and demographic stochasticity. In its original form, CATS minimizes the Kullback–Leibler divergence between the relative

Corresponding author
Zoltán Botta-Dukát, botta-dukat.zoltan@ecolres.hu

abundance expected a priori and the predicted relative abundances under the constraint that weighted trait means (CWMs) have to be equal in the predicted and observed communities. This approach uses observed trait means as the input and does not require the observed abundances (see its implementation in FD package; *Laliberté, Legendre & Shipley, 2014*). This fact suggests that the method could be applied to any type of abundance data if relative abundances can be calculated (*i.e.*, relative abundances are the weights in the CWM calculation).

*Warton, Shipley & Hastie (2015)* have shown that a Poisson-regression for abundances using species' traits as predictors and the logarithm of relative abundances expected a priori as an offset results in the same parameter estimates. The Poisson regression has three assumptions: (1) the abundance values follow a Poisson distribution; (2) the logarithm of expected abundances is a linear function of trait values; and (3) observations are independent (conditional on trait values). Violation of the third assumption has implications for interference (see *e.g.*, *Warton, Shipley & Hastie, 2015*; *Ter Braak, Peres-Neto & Dray, 2017* for discussion on interference, when independence assumtion is violated due to species interactions). However, it does not lead to biased estimates of model parameters. Violation of the first and second assumptions may result in biased parameter estimates.

The "examples" section will illustrate that violation of the Poisson assumption may lead to false conclusions.

A Poisson distribution of abundances can be assumed when they are measured by a number of individuals. Even in this case, the abundances may be over-dispersed (*i.e.*, the variance is higher than the mean), while a Poisson distribution implies equal mean and variance. Abundance is often measured in other units: estimated cover, biomass, and frequency of presence, and sometimes, only presence/absence data are available. Replacing the maximum entropy formalism with maximum likelihood (ML) fitting of generalized linear models (GLMs) allows for a generalization of CATS that relaxes the distribution assumption. Following *Warton, Shipley & Hastie (2015)*, we refer to GLMs for abundances with trait values as predictors as CATS regression. These models are not mathematically equivalent to CATS, but they have the same goal as CATS using distributional assumptions that better fit the abundance data at hand.

Depending on the unit of abundances, different distributions can be assumed. Presence/absence data can be modeled by logit-regression and assuming a binomial (Bernoulli) distribution (*Warton & Hui, 2010*). Assuming independent sampling points, the frequency of occurrence in sampling points (*e.g.*, pin points; *Goodall, 1952*) can also be modeled by a binomial distribution (see *Damgaard, 2008*; *Damgaard, 2009*) for relaxing of the independence assumtion). If the number of individuals counted in a sampling with fixed intensity (*e.g.*, fixed sampling area or trapping time), the simplest assumption is that abundances follow a Poisson distribution. However, abundance data may be over-dispersed (*i.e.*, the variance is higher than mean), zero-inflated (the number of zeros is higher than expected from the fitted distribution), or both.

Over-dispersed counts can be modeled by a negative binomial distribution (*O'Hara & Kotze, 2010*) or Conway-Maxwell-Poisson distribution (*Lynch, Thorson & Shelton, 2014*).

For count data with excessive zeros, zero-inflated or two-part (hurdle) models can be fitted (*Zuur et al., 2009*; *Blasco-Moreno et al., 2019*). If the total number of individuals is fixed in sampling instead of the sampling intensity, the number of individuals in each species follows a multinomial distribution (*Chong & Spencer, 2018*). The abundance of plant species is often described by their cover (which is often visually estimated). Cover data can be analyzed by (zero-inflated) beta-regression (*Damgaard & Irvine, 2019*). When individuals considerably differ in size, biomass may be a better abundance measure than the number of individuals. For modeling biomass data, a Tweedie distribution could be applied. It assumes that the mean-power relationship follows the Taylor law. If the power parameter p is in the range of $1<p<2$, a Tweedie distribution is mathematically equivalent to a compound Poisson-gamma distribution (*i.e.*, the sums of the Poisson-distributed number of individuals each have a gamma distributed mass). This distribution has a point mass at zero (*i.e.*, an absence of species) (*Dunstan et al., 2013*).

All of these distributions can be applied in generalized linear models. The interpretation of fitted parameters is similar for all distributions: a positive parameter value means that a higher trait value results in higher expected abundance. However, the relationship is nonlinear (except when applying an identity link) and depends on the applied link function. Therefore, plotting the expected abundances against traits gives a more detailed picture. For interference, the same procedures can be applied irrespective of the distribution (*Warton, Shipley & Hastie, 2015*; *Ter Braak, Peres-Neto & Dray, 2017*). Thus, at first glance, generalization of the CATS regression seems to be straightforward. However, there are two points that need more consideration: choosing/interpreting offset terms and calculating the explained variation. The aims of this paper are (1) to show that recommendations for the original CATS model should be reconsidered when a Poisson distribution is replaced by another distribution and (2) to give a general solution for this replacement and detailed recommendations for the most often used distributions.

## Modeling meta-community effect via offset

Beyond local trait selection, larger-scale effects can also influence species' local abundance. High propagule pressure can increase the local abundance of species that are abundant in the surroundings. On the other hand, locally well-adapted species may be missing from the local community due to propagule limitation. The unique property of CATS is that it can measure the relative importance of local and meta-community-scale (dispersal) processes (*Shipley, 2014*).

If species survival, growth, and reproduction were independent of their traits, local abundances would differ from the meta-community-level means due to demographic stochasticity only (including stochasticity of dispersal). In this case, local abundances could be predicted well from mean abundances at the meta-community level, while using traits as predictors would not improve the fit. At the other extreme, when species abundances are independent of dispersal processes (*i.e.*, no mass effect or propagule limitation) and fully determined by local processes, knowledge on the meta-community-level abundance would not be able to improve our ability to predict local abundances (*Shipley, 2014*). Therefore, the heart of CATS is the fitting of models with and without information on abundances

in the meta-community level. *Shipley (2014)* called this information the "neutral prior." However, *Warton, Shipley & Hastie (2015)* called attention to the term "prior" as being associated with Bayesian statistics, where it has a specific meaning. To avoid confusion, we will refer to it as "abundances expected *a priori*," where "*a priori*" means "before knowing local conditions."

The abundances expected a priori can be included into CATS regression models via offset terms. *Warton, Shipley & Hastie (2015)* suggested using the logarithm of relative abundances at the meta-community level in a Poisson regression to reproduce the original CATS model. This study discusses whether this suggestion is generally valid irrespective of the link function.

The aim of CATS models is to predict relative abundances of species, not to explain differences in total abundances among sites, which may be caused by differences in sampling intensity. Therefore, they always contain an intercept. The relative abundance predicted by a model containing only an intercept and offset should be equal to the relative abundances expected a priori ($\pi$):

$$\frac{\hat{y}_i}{\sum_{i=1}^{S}\hat{y}_i} = \pi_i \tag{1}$$

where $\hat{y}_i$ is the predicted abundance of species $i$, $S$ is the number of species, and $\pi_i$ is relative abundances expected a priori of species $i$. For most of the fitted models (but not for zero-inflated and two-stage models), $\hat{y}_i = \mu_i$, where $\mu_i$ is the location parameter of the fitted distribution. Therefore, $\mu_i$ will be used instead of $\hat{y}_i$ where appropriate. If a canonical link is applied and the model contains an intercept term, the sum of predicted values is equal to the sum of observed values ($y_{tot}$). Thus, requirement Eq. (1) could be written in the following form:

$$\mu_i = \pi_i y_{tot}. \tag{2}$$

The GLM with an intercept and offset but no predictors can be written in the following general form:

$$h(\mu_i) = \beta_0 + O_i \tag{3}$$

where h() is the link function, $\beta_0$ is the intercept, and $O_i$ is the offset for species $i$. Substituting Eq. (2) into Eq. (3), we obtain the following system of linear equations (note that the left side of equations can be replaced by numbers calculated from data on hand):

$$h(\pi_i y_{tot}) = \beta_0 + O_i. \tag{4}$$

This system contains $S + 1$ variables ($O_1, O_2, \ldots, O_S$ and $\beta_0$), but only $S$ equations, so it has no unique solution. It can be solved by choosing an arbitrary value for $O_1$. Then, the other offsets can be calculated with the following formula:

$$O_i = h(\pi_i y_{tot}) - h(\pi_1 y_{tot}) + O_1. \tag{5}$$

In Poisson and negative binomial regression, the canonical link is the natural logarithm (*Dobson, 2002*). Thus,

$$O_i = ln(\pi_i y_{tot}) - ln(\pi_1 y_{tot}) + O_1 = ln(\pi_i) - ln(\pi_1) + O_1. \tag{6}$$
In this case, it is appropriate to choose $O_1 = ln(\pi_1)$, which leads to the offset recommended by *Warton, Shipley & Hastie (2015)*. For other link functions, the simplest choice is $O_1 = 0$. However, this approach has two limitations. First, $\pi_i y_{tot}$ has to be within the domain of the link function. For example, for binomial and beta distributions, where the canonical link is logit$(x)$, offsets can be calculated only if $\pi_i y_{tot} < 1$; otherwise, $ln(\pi_i y_{tot}/(1 - \pi_i y_{tot}))$ cannot be calculated. The second limitation is that a canonical link is not always the most appropriate link function, and sometimes, another link function has to be chosen. For example, for a Tweedie distribution with power parameter 1<p<2, the canonical link would be (*Ohlsson & Johansson, 2006*):

$$h(\mu) = \frac{-1}{p-1}\mu^{-(p-1)}. \tag{7}$$

When the exact value of power parameter $p$ is unknown, it can be estimated from data during model fitting, but in this case, the log-link is applied in R packages mgcv (*Wood, 2017*) and glmmTMB (*Brooks et al., 2017*). If a canonical link is not used, the sum of expected values may differ from the sum of observed values. But Eq. (1) remains true irrespective of the link function, and it can be converted to:

$$\frac{\mu_i}{\mu_1} = \frac{\pi_i}{\pi_1}. \tag{8}$$

Combining Eqs. (3) and (8) results in:

$$\frac{h^{-1}(\beta + O_i)}{h^{-1}(\beta + O_1)} = \frac{\pi_i}{\pi_1} \tag{9}$$

where $h^{-1}(x)$ is the inverse of link function.

Setting $O_1$ to an arbitrary value leads to a nonlinear equation system with $S$ equations and $S$ variables. Solving such a system is often a hard task. The situation would be much simpler if the following were true:

$$h^{-1}(\beta + O_i) = h^{-1}(\beta)h^{-1}(O_i). \tag{10}$$

In this case, choosing $O_i = h(\pi_i)$ satisfies condition Eq. (9). Since $h(x) = ln(x)$ and $h^{-1}(x) = e^x$ satisfy condition Eq. (10), it is reasonable to use log-link even if it is not the canonical link (for example, instead of logit in a binomial model). In medical statistics, binomial GLM with log-link is called relative risk regression, which is often recommended due to the easier interpretation of proportions than odds ratios (*Marschner, 2015*). If log-link is used with a binomial distribution, iteratively reweighted least squares (the standard method for fitting GLMs) may fail to converge to the maximum likelihood estimate (*Marschner & Gillett, 2012*). Therefore, alternative estimation procedures were developed and are implemented in the *logbin* R package (*Donoghoe & Marschner, 2018*). Log-link for beta regression is also available in the *betareg* R package (*Cribari-Neto & Zeileis, 2010*).

Using the mentioned distributions, we suppose that all data come from the same distribution, and only their parameters depend on species. In this case, the relative abundances at the meta-community level can be estimated by:

$$\pi_i = \frac{\overline{m_i}}{\sum \overline{m_i}} \tag{11}$$
where $\overline{m_i}$ is the mean abundance of species $i$ in plots representing the meta-community. Note that since only the ratio of relative abundances expected a priori are used, the offset could be simply $O_i = ln(\overline{m_i})$ instead of $O_i = ln(\pi_i)$.

If there are excessive zeros, it could be supposed that some of the zeros do not come from the distribution of "normal" abundances. Zero-inflated and two-part (hurdle) models are based on this assumption. In these models, there are two equations for two location-type parameters.

A zero-inflated model supposes that positive counts and some of the zeros comes from a Poisson or negative binomial process, while some zeros are "structural zeros" (*i.e.*, species cannot occur there). The two parameters in this case are the probability of structural zeros ($p$) and the expected value of the Poisson or negative binomial process ($\mu$) (*Zuur et al., 2009*). For estimating offsets, we must know the probability of structural zeros at the meta-community. Therefore, it seems that zero-inflated models have low practical relevance when a meta-community effect has to be modeled.

A two-part (hurdle) model fits two separate models: a binomial model for presence/absence data and a truncated Poisson or negative binomial model for positive abundances. In this case, the two parameters are the probability of presence ($p$) and mean of the Poisson or negative binomial distribution ($\mu$), from which the fitted zero-truncated distribution is deduced (not the mean of the truncated distribution itself) (*Zuur et al., 2009*). For simplicity, let us imagine that we really fit a two-part model as two separate GLMs. The first GLM is a binomial model for binary data. Therefore, we estimate offsets using the standard procedure (but mean abundances at the meta-community level have to be calculated from binary data). In the second GLM, we fit a truncated Poisson or truncated negative binomial distribution for the non-zero abundances. Similar to Poisson regression, it is assumed that $ln(\mu_i)$ is a linear combination of trait values and the offset, but the expected abundance is the following for a Poisson distribution:

$$\hat{y}_i = \frac{\mu_i}{1 - \exp(-\mu_i)}. \tag{12}$$

For a negative binomial distribution, the expected abundance is:

$$\hat{y}_i = \frac{\mu_i}{1 - \left(\frac{\mu_i + \theta}{\theta}\right)^{-\theta}} \tag{13}$$

$\hat{y}_i$ is the expected or mean abundance of species $i$ when it present, while $\mu_i$ is the expected or mean abundance when only structural zeros are excluded. For setting the offset, we need a priori expectation for the latter.

The meta-community level mean of species' abundance when present ($\overline{m}_i^+$) can be easily estimated. Assuming a Poisson process, the mean abundance excluding structural zeros ($\tilde{m}_i$) can be estimated by solving the following nonlinear equation:

$$\overline{m}_i^+ = \frac{\tilde{m}_i}{1 - \exp(-\tilde{m}_i)} \tag{14}$$

If a negative binomial distribution is assumed, a similar approach can be applied if $\theta$ is known. Then, $ln(\tilde{m}_i)$ could be used as an offset.

## Relative importance of environmental selection and dispersal processes

The relative importance of local and meta-community-level processes can be calculated from variation explained by models containing only traits (as independent variables), only offset (calculated from meta-community-level abundances; see above), or both traits and offset (*Shipley, 2014*). Using the classic $R^2$ as a measure of explained variance is suitable only in OLS regression. Different generalizations of $R^2$ are suggested for GLMs (*Cameron & Windmeijer, 1996*; *Cameron & Windmeijer, 1997*; *Menard, 2000*; *Nakagawa & Schielzeth, 2013*). *Shipley (2014)* proposed using a generalization based on Kullback–Leibler divergence (*Cameron & Windmeijer, 1997*).

I will show below that formula Eq. (4) from *Shipley (2014)* is valid for only a Poisson-model, and different formulas have to be used for other distributions. A definition of Kullback–Leibler $R^2$ is not available for models with an offset; therefore, solution for this case by *Shipley (2014)* and its alternatives will be discussed. Increasing the number of predictors (traits) always improves the fit of the model (*i.e.*, increases the Kullback–Leibler $R^2$). Therefore, $R^2$ values of models with different numbers of predictors (traits) cannot be compared. *Shipley (2014)* proposed an "adjustment" procedure based on randomization of traits. Although this procedure is correct, it is time consuming for large datasets . Thus, an alternative deterministic adjustment is proposed.

## $R^2$ for models without offset

Kullback–Leibler $R^2$ is a generalization of the classic $R^2$ used in ordinary least squares regression:

$$R^2 = 1 - \frac{\sum (y_i - \hat{y}_i)^2}{\sum (y_i - \overline{y})^2} = \frac{\sum (y_i - \overline{y})^2 - \sum (y_i - \hat{y}_i)^2}{\sum (y_i - \overline{y})^2} \tag{15}$$

where $\sum (y_i - \hat{y}_i)^2$ and $\sum (y_i - \overline{y})^2$ are the squared Euclidean distances between observed values and predictions of models with and without predictors, respectively. Therefore, $R^2$ is a proportional decrease of distance between model prediction and observed values due to the inclusion of predictors in the model. For other distributions, the squared Euclidean distance can be replaced with Kullback–Leibler divergence with the same interpretation:

$$R_{KL}^2 = 1 - \frac{K(\mathbf{y}; \boldsymbol{\mu})}{K(\mathbf{y}; \boldsymbol{\mu}^0)} = \frac{K(\mathbf{y}; \boldsymbol{\mu}^0) - K(\mathbf{y}; \boldsymbol{\mu})}{K(\mathbf{y}; \boldsymbol{\mu}^0)} \tag{16}$$

where $\mathbf{y}$ is the vector of observed values, and $\boldsymbol{\mu}$ and $\boldsymbol{\mu}^0$ are vectors of values predicted by the evaluated and intercept-only models, respectively.

Table 1 shows that distributions that could be used in CATS regression belong to the exponential family, so their density functions can be written in the following general form (*McCullagh & Nelder, 1999*):

$$f(y; \psi, \phi) = \exp \left\{ \frac{y\psi - b(\psi)}{a(\phi)} - c(y, \phi) \right\} \tag{17}$$

**Table 1** **Defining distributions widely used for modeling abundances using notations of exponential family.** See Eq. (17) for explanation of notations.

| Distribution | $\psi$ | $b(\psi)$ | $a(\phi)$ | $c(y,\phi)$ |
|---|---|---|---|---|
| Gaussian (Normal) | $\mu$ | $\frac{\psi^2}{2}$ | $\sigma^2$ | $\frac{\sigma^2 \ln(2\pi\sigma^2)+y^2}{2\sigma^2}$ |
| Poisson | $\ln\mu$ | $e^\psi$ | 1 | $-\ln y!$ |
| Binomial | $\ln\frac{\mu}{n-\mu}$ | $n\ln(1+e^\psi)$ | 1 | $\ln\binom{n}{y}$ |
| Negative binomial | $\ln\frac{\mu}{\mu+\theta}$ | $-\theta \ln(1-e^\psi)$ | 1 | $\ln\frac{\Gamma(\theta)y!}{\Gamma(y+\theta)}$ |
| Tweedie($1<p<2$) | $\frac{\mu^{1-p}}{1-p}$ | $\frac{[\psi(1-p)]^{(2-p)/(1-p)}}{2-p}$ | $\phi$ | $0$ if $y=0$ <br> $\ln W(y,0)-\ln y$ if $y>0$ |
| Zero-truncated Poisson | $\ln\mu$ | $e^\psi+\ln[1-exp(-e^\psi)]$ | 1 | $-\ln y!$ |
| Zero-truncated negative binomial | $\ln\frac{\mu}{\mu+\theta}$ | $-\theta \ln(1-e^\psi)+\ln[1-(1-e^\psi)^{-\theta}]$ | 1 | $\ln\frac{\Gamma(\theta)y!}{\Gamma(y+\theta)}$ |

**Notes.**

Notation: $\Gamma(x)$ is the gamma-function.

where $\psi$ is the natural or canonical parameter, $\phi$ is the dispersal parameter, and a, b, and c are specific functions. The mean and variance of y are:

$$E(\mathbf{y})=\boldsymbol{\mu}=b'(\psi)$$
$$Var(\mathbf{y})=a(\phi)b''(\psi) \tag{18}$$

where $b'$ and $b''$ are the first and second derivatives of function $b$. Note that negative binomial and Tweedie distributions belong to this family only if parameters $\theta$ and $p$ are known constants. Function $h(\mu)=\psi$ is called a canonical link function.

For members of the exponential family, Kullback–Leibler divergence can be calculated as the difference between the likelihood of a full model (*i.e.*, a model where predicted and observed values are equal) and a fitted model (*Cameron & Windmeijer, 1997*):

$$K(\mathbf{y};\boldsymbol{\mu})=2\left[l\left(\boldsymbol{\mu}^{full};\mathbf{y}\right)-l(\boldsymbol{\mu};\mathbf{y})\right]. \tag{19}$$

Thus, for members of the exponential family, $R^2_{KL}$ could be deduced as a corrected version of likelihood ratio $R^2$ or McFadden $R^2$:

$$R^2_L = 1-\frac{l(\boldsymbol{\mu};\mathbf{y})}{l(\boldsymbol{\mu}^0;\mathbf{y})}. \tag{20}$$

A drawback of $R^2_L$ is that its maximum is not 1, but $1-\left[l\left(\boldsymbol{\mu}^{full};\mathbf{y}\right)/l\left(\boldsymbol{\mu}^0;\mathbf{y}\right)\right]$. Since its minimum is zero, $R^2_L$ can be rescaled to the interval of 0–1 by dividing it by its maximum, which results in $R^2_{KL}$:

$$\frac{R^2_L}{\left(1-\frac{l(\boldsymbol{\mu}^{full};\mathbf{y})}{l(\boldsymbol{\mu}^0;\mathbf{y})}\right)}=\frac{l(\boldsymbol{\mu}^0;\mathbf{y})-l(\boldsymbol{\mu};\mathbf{y})}{l(\boldsymbol{\mu}^0;\mathbf{y})}\Bigg/\frac{l(\boldsymbol{\mu}^0;\mathbf{y})-l(\boldsymbol{\mu}^{full};\mathbf{y})}{l(\boldsymbol{\mu}^0;\mathbf{y})}$$
$$=\frac{l(\boldsymbol{\mu}^0;\mathbf{y})-l(\boldsymbol{\mu};\mathbf{y})}{l(\boldsymbol{\mu}^0;\mathbf{y})-l(\boldsymbol{\mu}^{full};\mathbf{y})}=R^2_{KL}. \tag{21}$$

Substituting Eqs. (19) into (16) results in:

$$R^2_{KL} = 1 - \frac{l\left(\boldsymbol{\mu}^{full};\mathbf{y}\right) - l\left(\boldsymbol{\mu};\mathbf{y}\right)}{l\left(\boldsymbol{\mu}^{full};\mathbf{y}\right) - l\left(\boldsymbol{\mu}^0;\mathbf{y}\right)} = \frac{l\left(\boldsymbol{\mu};\mathbf{y}\right) - l\left(\boldsymbol{\mu}^0;\mathbf{y}\right)}{l\left(\boldsymbol{\mu}^{full};\mathbf{y}\right) - l\left(\boldsymbol{\mu}^0;\mathbf{y}\right)}. \tag{22}$$

*McCullagh & Nelder (1999)* called $2\left[l\left(\boldsymbol{\mu}^{full};\mathbf{y}\right) - l\left(\boldsymbol{\mu};\mathbf{y}\right)\right]$ the scaled deviance ($D^*$), so $R^2_{KL}$ can also be calculated from scaled deviances ($D^*$) or deviances ($D$) of fitted and intercept-only models:

$$R^2_{KL} = 1 - \frac{D^*\left(\boldsymbol{\mu};\mathbf{y}\right)}{D^*\left(\boldsymbol{\mu}^0;\mathbf{y}\right)} = 1 - \frac{D\left(\boldsymbol{\mu};\mathbf{y}\right)/a(\phi)}{D\left(\boldsymbol{\mu}^0;\mathbf{y}\right)/a(\phi)} = 1 - \frac{D\left(\boldsymbol{\mu};\mathbf{y}\right)}{D\left(\boldsymbol{\mu}^0;\mathbf{y}\right)}. \tag{23}$$

If there is no offset, in a generalized linear model fitted by ML estimation with a canonical link, the expectations in an intercept-only model is equal to the mean of observed values: $\mu^0 = \overline{y}$. Formulas for this case are listed in Appendix S1. Appendix S2 shows that formula Eq. (4) from *Shipley (2014)* is equivalent to the formula given for a Poisson regression in Appendix S1.

These formulas assume that the likelihood is a function of $\mu$ only, and if there are other parameters, their values are constants known a priori (*i.e.*, not estimated during regression). If these parameters are estimated in regression, we can obtain different estimates for the evaluated and the intercept-only models. For a negative binomial distribution with unknown dispersion ($\theta$), *Cameron & Windmeijer (1996)* suggested using a parameter estimated for an evaluated model when the likelihood of full and intercept-only models is calculated. Applying this approach, $R^2_{KL}$ may decrease when a regressor is added to the model due to changes in estimated $\theta$. The same approach can be applied for the power parameter of a Tweedie distribution. GLM fitting programs usually give the log-likelihood of the fitted model and the deviance of fitted and intercept-only models.

## R² for models with offset

In the previous section, $\boldsymbol{\mu}$ was the prediction of model containing an intercept and predictors (traits), while $\boldsymbol{\mu}^0$ was the prediction of an intercept-only model. A possible solution for how we should include the offset is to define $\boldsymbol{\mu}^0$ as the prediction of a model without predictors (*i.e.*, a model containing only an intercept and offset). At first glance, it seems to be a natural generalization of R² shown in the previous section. However, there is a drawback in this approach: the effect of meta-community-level processes cannot be directly measured since R² for models containing only an intercept and offset (but no traits) would always be zero. *Shipley (2014)* suggested an indirect measure of a pure meta-community effect:

$$\frac{R^2\left(\text{traits; offset}\right) - R^2\left(\text{traits}\right)}{1 - \overline{R^2\left(\text{random traits}\right)}}. \tag{24}$$

The nominator is an adjustment for removing bias (see next section on adjustment), so now, we should focus on the denominator. This subtraction is based on the assumption that $R^2(\text{traits})$ is the variation explained by traits, while $R^2(\text{traits; offset})$ is the variation explained by traits and offset (*i.e.*, meta-community effect) together. However, this assumption is not satisfied when in calculation of R-squared $\boldsymbol{\mu}^0$ is the prediction of a
model containing only an intercept and offset. To understand why, we should recall the geometric interpretation of Kullback–Leibler $R^2$ shortly mentioned above: $R^2$ is the proportional decrease of distance between observed and predicted values (or proportional improvement of fit) due to the inclusion of predictors in the model. Therefore, in Eq. (24), both $R^2$(traits) and $R^2$(traits; offset) are proportional improvements of fit due to the inclusion of traits, but they are proportional to different original distances of observed and predicted values. Therefore, their difference has no simple interpretation and does not measure the pure meta-community effect.

Instead of defining $\mu^0$ as a prediction of a model without a predictor, it could be defined as a prediction of an intercept-only model, even if an offset is applied. This definition allows us to calculate a meaningful $R^2$ for models with offset but no predictors as a direct measure of the meta-community effect. This definition may result in negative $R^2$ when including an offset increases the distance between observed and predicted values instead of decreasing it. A negative value is nonsense if $R^2$ is interpreted as explained variation, but it is meaningful if $R^2$ is interpreted as a proportional change in the distance between observed and predicted values. This geometric interpretation seems more useful in CATS regression, where it has a meaning that includes meta-community-level relative abundances decreases the goodness-of-fit.

Note that in R environment, to avoid negative $R^2$ values, the following formula is applied instead of Eq. (15):

$$R^2 = 1 - \frac{\sum (y_i - \hat{y}_i)^2}{\sum (y_i - \hat{y}_i)^2 + \sum (\hat{y}_i - \bar{y})^2}. \tag{25}$$

If there is no offset, Eqs. (15) and (25) result in the same value, but they differ if offset is applied. A generalization of Eq. (25) could be:

$$R^2 = 1 - \frac{K(\mathbf{y}; \boldsymbol{\mu})}{K(\mathbf{y}; \boldsymbol{\mu}) + K(\boldsymbol{\mu}; \boldsymbol{\mu}^0)}. \tag{26}$$

## Adjusted $R^2$ and partitioning of explained variation

As a goodness-of-fit measure, a drawback of $R^2$ (both in the classic form and its generalization) is that including an additional predictor in the model always increases $R^2$, even if the predictor is independent of the dependent variable. Therefore, it has a positive expected value instead of zero when a dependent variable is not related to the predictors. To remove this bias, *Fisher (1925)* suggested using the following adjustment:

$$R^2_{adj} = 1 - \left(1 - R^2\right) \frac{n-1}{n-k-1} \tag{27}$$

where $n$ is the number of data points, and $k$ is the number of predictors.

Unfortunately, this adjustment is valid for only for "classic" $R^2$ of ordinary least squares regression. Since *Shipley (2014)* has not found a similar solution for Kullback–Leibler $R^2$, he proposed a procedure based on reshuffling trait values to remove the bias. Although the suggested procedure is correct, it has no unique result, in contrast to the correction using a

closed form. *Ricci (2010)* has shown that there is a simple general adjustment for $R_{KL}^2$ when the dependent variable follows a distribution belonging to the exponential family. Let us write the formula for $R_{KL}^2$ using scaled deviances:

$$R_{KL}^2 = \frac{D(\boldsymbol{\mu}^0; \mathbf{y}) - D(\boldsymbol{\mu}; \mathbf{y})}{D(\boldsymbol{\mu}^0; \mathbf{y})}. \tag{28}$$

Recall that $R_{KL}^2$ is the proportional improvement of model fit. Let us focus on the denominator, which is the absolute improvement. In a bias-corrected version, the absolute improvement should be zero when predictors have no effect on the dependent variable. Under this condition, for members of the exponential family, $D(\boldsymbol{\mu}^0; \mathbf{y}) - D(\boldsymbol{\mu}; \mathbf{y})$ approximately follows a Chi-square distribution with degree of freedom equals to the number of predictors $(k)$. Since the expected value of the Chi-square distribution is its degree of freedom, the following is an approximately bias-free goodness-of-fit measure:

$$R_{KL,adj}^2 = \frac{D(\boldsymbol{\mu}^0; \mathbf{y}) - D(\boldsymbol{\mu}; \mathbf{y}) - k}{D(\boldsymbol{\mu}^0; \mathbf{y})}. \tag{29}$$

Appendix S3 shows that Eq. (27) is a special case of Eq. (29) for a Gaussian distribution with dispersion estimated from the data. Note that *Ricci (2010)* applied an alternative derivation of adjustment by generalization of the shrinkage factor and used the deviance instead of the scaled deviance in the formulas.

For partitioning variation, we should fit models containing both traits and offset and models with only traits and only offset. Let us denote the corresponding adjusted $R^2$ values by $R^2$(trait; offset), $R^2$(trait), and $R^2$(offset), respectively. $R^2$(trait; offset) measures the whole variation explained by studied traits and relative abundances at the meta-community level. The pure trait effect (*i.e.*, variation explained only by traits) is $R^2$(trait; offset)-$R^2$(offset), while the pure meta-community effect is $R^2$(trait; offset)- $R^2$(trait). Variation that can be explained by both traits and the meta-community effect is $R^2$(trait)+$R^2$(offset)-$R^2$(trait; offset). These formulas are analogous to partitioning of the variation of community composition into environmental and spatial components (*Borcard, Legendre & Drapeau, 1992*; *Peres-Neto et al., 2006*).

# EXAMPLES

Examples are presented to illustrate the main messages of the paper. First, example 1 shows how setting an inappropriate distribution leads to biased parameter estimates. Next, Example 2 shows that it is important to choose an appropriate offset, and finally Example 3 illustrates why variation components should be estimated in a new way.

Examples uses new R package *CATSregression* publicly available on GitHub (https://github.com/BottaDZ/CATSregression/). The package's vignette shows more examples using field data.

## Example 1: fitting Poisson model to over-dispersed counts

The first example illustrates the problems that arise when an inappropriate model is fitted. The type of abundance often clearly determines the type of model to be fitted. However,

when abundance is measured by a number of individuals, a Poisson or negative binomial model should be fitted depending on whether there is a random or aggregated spatial pattern of individuals. If the spatial pattern is random, the number of individuals will follow a Poisson distribution, where the variance is equal to the mean. An aggregated spatial pattern leads to over-dispersed counts (*i.e.*, the variance is higher than mean), which can be modeled by a negative binomial distribution.

The presence of over-dispersion can be checked by comparing the Akaike Information Criteria of Poisson and negative binomial models or using diagnostic plots of residuals (Fig. 1). Dunn-Smyth (or randomized quantile) residuals (*Dunn & Smyth, 1996*) are especially useful for this purpose because if the model's assumptions (specified distribution and log-linear relationship) are satisfied, they follow a standard normal distribution (*Warton, Shipley & Hastie, 2015*; *Feng, Li & Sadeghpour, 2020*).

This example uses simulated data. Abundances (y) of 20 species in a plot were simulated. Abundances follow a negative binomial distribution with a mean that has a log-linear relation to values of a trait. The dispersion parameter is constant, and trait values follow a normal distribution:

$$y_i \sim \text{NegBin}\left(\mu_i = \exp\left(0.5 * x_i\right), \theta = 1\right)$$
$$x_i \sim N\left(m = 10, \sigma = 3\right).$$

(30)

The simulation was repeated 50 times, and Poisson and negative binomial models were fitted to each simulated plots separately. Figure 1 shows a typical diagnostic plot, which has a strong nonlinearity of the QQ plot indicating that the distributional assumption of a Poisson regression is not satisfied. The diagnostic plots of a negative binomial model do not indicate any problem.

The estimated slopes are dispersed around the real value (0.5) in both Poisson and negative binomial models (Fig. 2). The variation among estimates was higher in the Poisson distribution, while the confidence intervals were narrower due to the neglect of over-dispersion. These two facts together may result in over-interpretation of differences in the strength of selection among plots.

## Example 2: choosing appropriate offset

The second example illustrates the importance of choosing an appropriate offset, without which fitted relative abundances may differ considerably from the relative abundances in the meta-community, even if no traits are included in the model. The example uses the dataset of *Raevel, Violle & Munoz (2012)*, which contains the abundance (number of individuals) of 97 species at 52 sites. Data were transformed into a presence/absence scale, and then meta-community-level abundances were measured as the number of occurrences at the 52 sites.

For modeling of the presence/absence data, a binomial distribution has to be applied. The canonical link for this distribution is the logit link. The proposed way of calculating the offset for the logit link in this paper cannot be applied because the product of the number of species in the plot and relative abundance at the meta-community scale was larger than one in 39 species-site combinations (and it excludes 22 of 52 sites). Therefore, the logarithm of relative abundances in the meta-community was used.

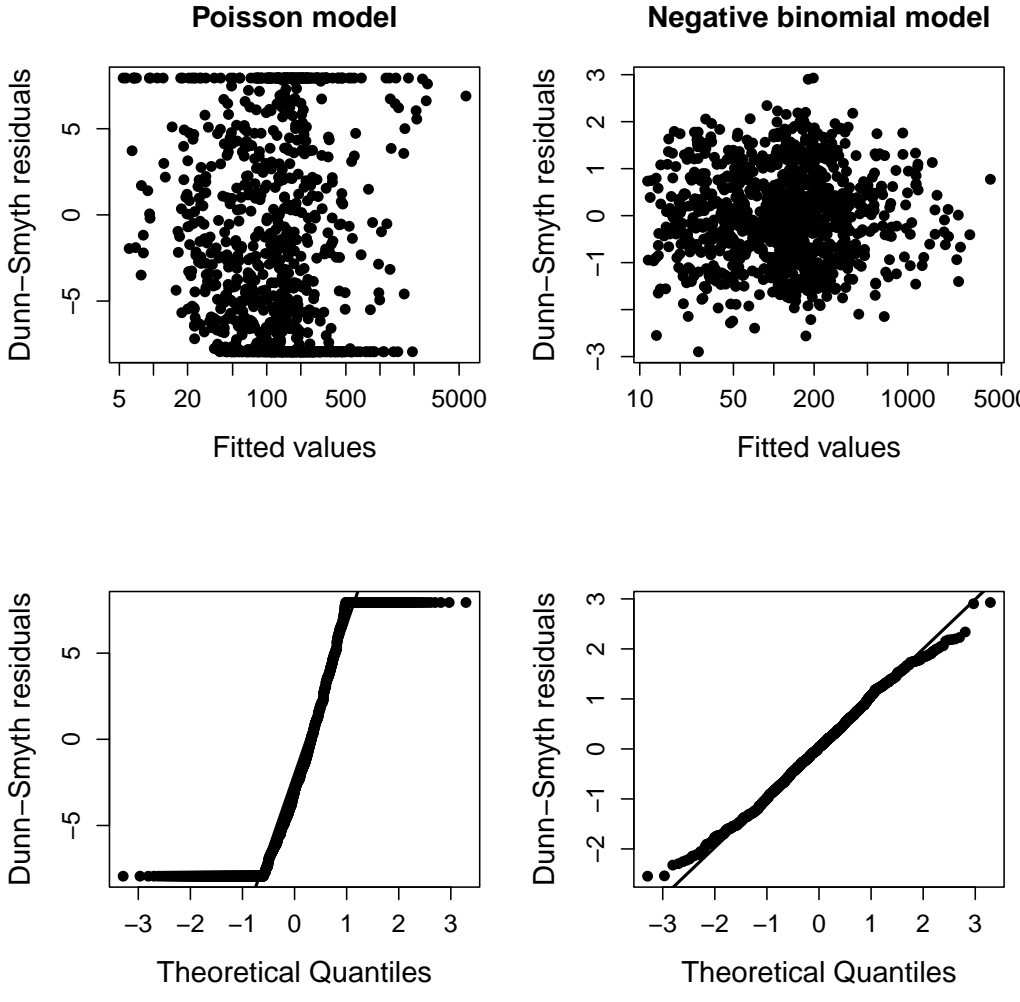

**Figure 1** **Diagnostic plots of models fitted to one community of Example 1.** The fan shape of points in residuals vs. fitted values plot (upper row) and departure from the expected line in QQ-plot (bottom row) indicate that Poisson model is inappropriate due to over-dispersion.

Two link functions were tested: the canonical (logit) link and log link, as suggested in this paper. The latter was fitted using the *logbin* package (*Donoghoe & Marschner, 2018*).

A model containing only intercept and offset terms was fitted, so the predicted relative abundances in plots should equal to the relative abundance in the meta-community. This requirement was satisfied in the model using log link (not shown). However, when the canonical link was used, there is a non-linear relationship between two vectors of relative abundances (Fig. 3).

## Example 3: comparing formulas for estimation of variation components

The aim of the next example is to compare variation components estimated by the method of *Shipley (2014)* and the new method proposed in this paper. Simulated data were used, where trends of variation components were predictable. Data were generated using the
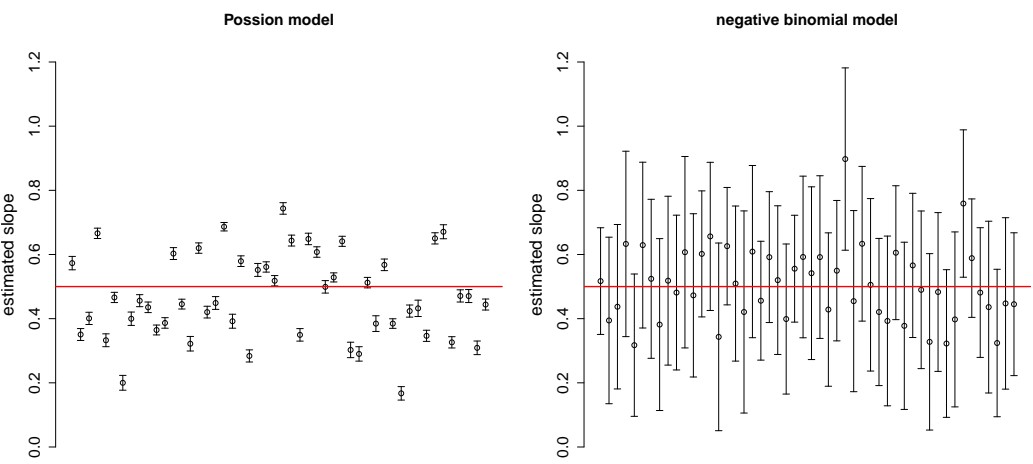

**Figure 2** **Estimated slopes with their 95% confidence intervals in 50 simulated plots of Example 1.** Red horizontal line indicates the real slope used in the simulation.

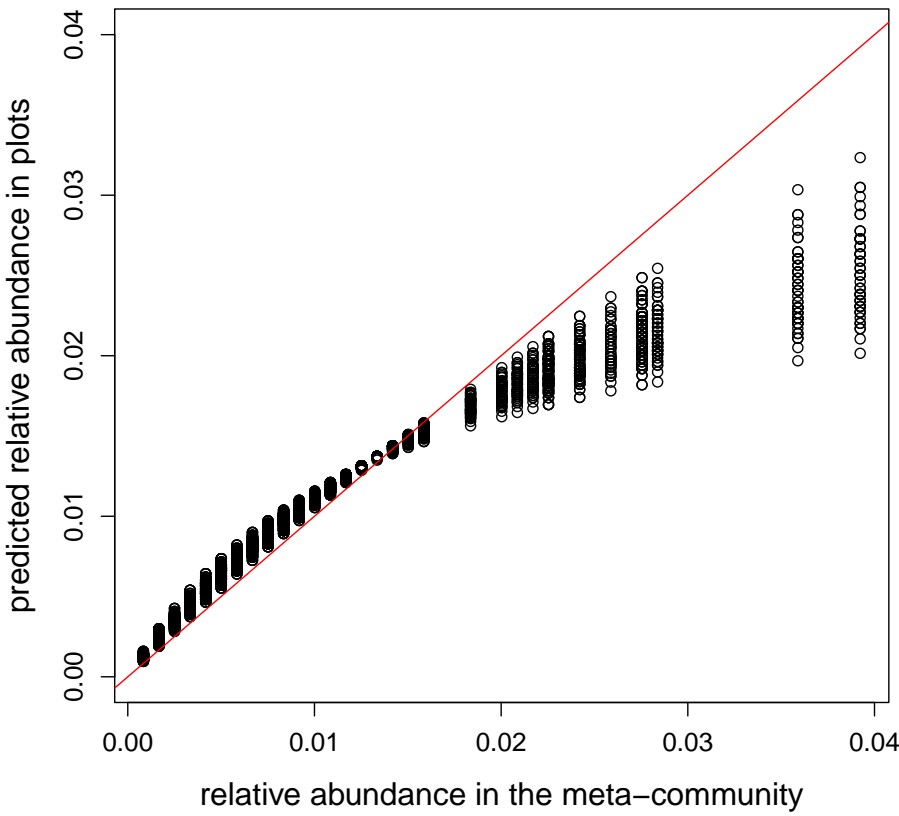

**Figure 3** **Relationship between meta-community level and predicted relative abundances in model without traits using logit link.** Since local selection is not modelled, points should lie the red 1:1 line.

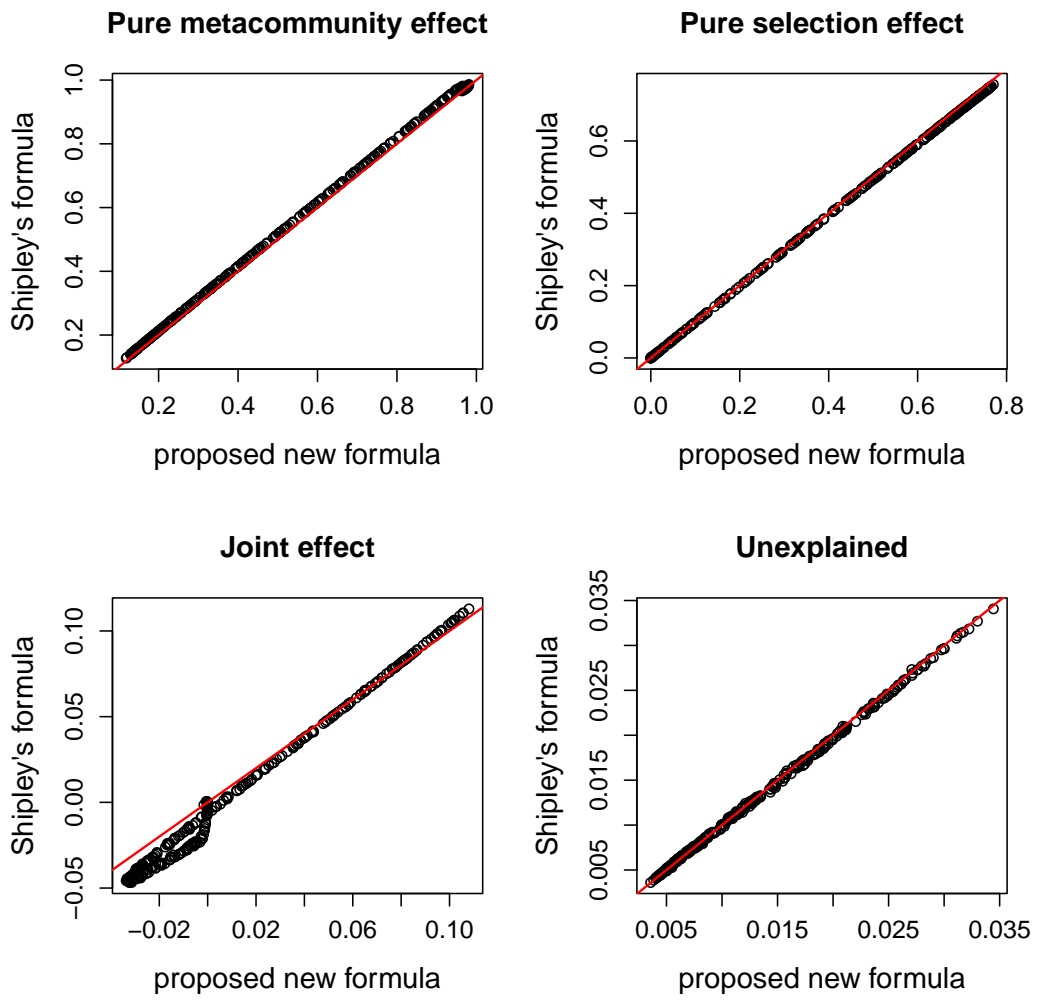

**Figure 4  Comparing variation components calculated by Shipley's formulas and new formulas proposed in this paper.** Components calculated by two ways show good agreement.

following model:

$$y_i \sim \text{Poisson}(\lambda_i)$$
$$log\,\lambda_i = a + log\,\pi_i + s * t_i \qquad (31)$$

where $\pi_i$ is the meta-community-level relative abundances, $|s|$ is the strength of selection, and $t_i$ is the trait value. To remove changes due to total community size, the intercept (a) was set to:

$$a = logA - log\sum_i \pi_i e^{st_i}. \qquad (32)$$

Thus, $\sum \lambda_i = A$ for any value of $s$.

The species pool consists of 50 species, and their traits follow a standard normal distribution. The expected community size (A) was set to 2500, and the strength of selection (s) changes from 0 to 3. Pure selection and pure meta-community effects were

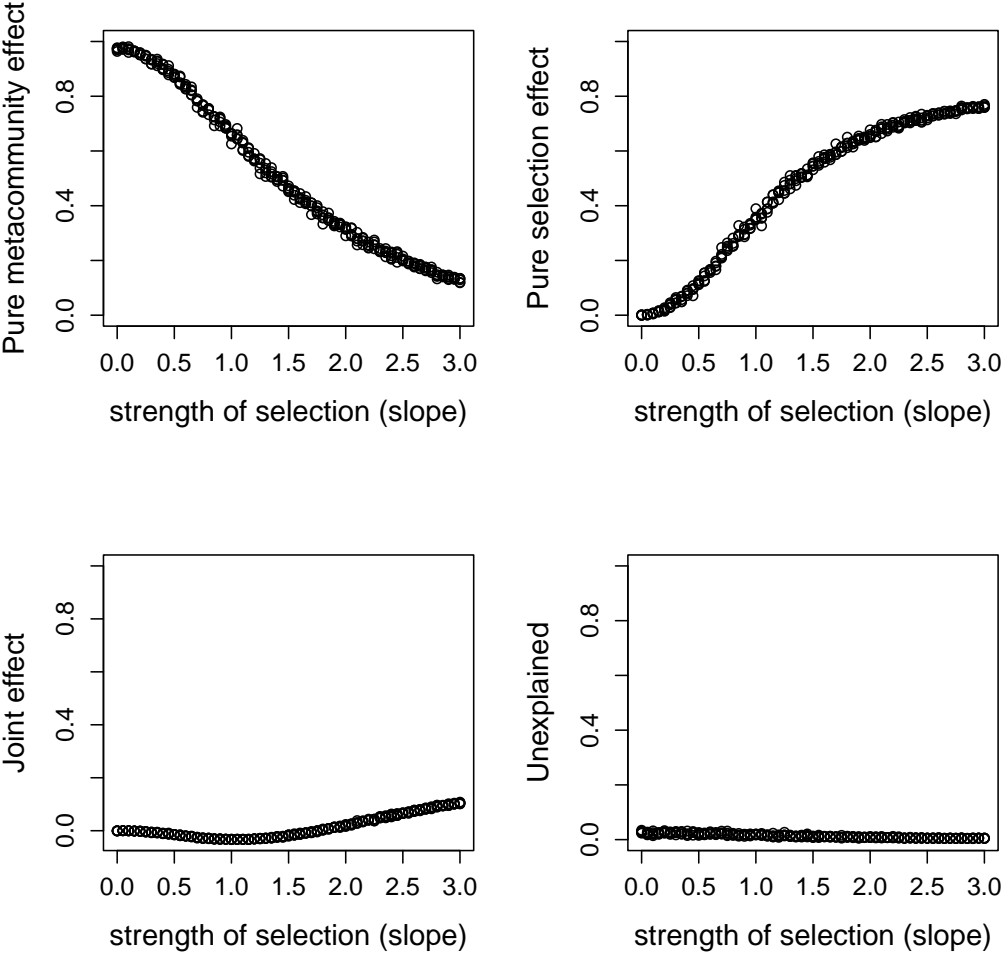

**Figure 5** Variation components in simulated communities that differ in strength of selection calculated by formulas proposed in this paper. As expected, meta-community effect decreases, while selection effect increases with increasing strength of selection, and the former is near zero at $s = 0$ when there is no selection in the simulation.

calculated for each simulated community separately using formulas from *Shipley (2014)* and the method proposed in this paper.

There is a good agreement between variation components calculated by the two ways (Fig. 4). As expected, the pure meta-community effect decreases, while the pure selection effect increases with increasing strength of selection, and the later was about zero at $s = 0$ (Fig. 5). The joint effect and unexplained variation do not change considerably with changing strength of selection.

## CONCLUSIONS

CATS regression is a useful part of community ecologists' toolbox to understand how the environment selects species through trait-environment relationships, as well as to estimate the relative role of local environmental selection and meta-community-level processes in

the assembly of communities. In its original version using maximum entropy formalism (*Shipley, Vile & Garnier, 2006*; *Shipley, 2010*), the assumptions of the methods remain hidden. Converting the maximum entropy formalism into a GLM (*Warton, Shipley & Hastie, 2015*) made the assumptions explicit. *Warton, Shipley & Hastie (2015)* focused on the most important assumption (the distribution of abundance values) and the central part of the method (the estimation of parameters). The original version assumes a Poisson distribution, and not only parameter estimates, but also the additional parts of the method (defining offset terms and calculating R-squared values) may change when data follow an other distribution.

Parameter estimation for different distributions is a well-known statistical problem, and a user can easily choose the appropriate function (or option of the applied function). This paper focused on additional parts of the method, which are more specific and have thus received little attention so far. Theoretical considerations and examples illustrated that naively using algorithms developed for a Poisson distribution may be misleading when data follow other distributions. The recommendations formulated in this paper could help to avoid these potential pitfalls.

## ACKNOWLEDGEMENTS

Thanks to Bill Shipley and Nigel Yoccoz for their helpful comments.

### Funding
This project was supported by the National Research, Development and Innovation Office of Hungary (No, 124671). The funders had no role in study design, data collection and analysis, decision to publish, or preparation of the manuscript.

### Grant Disclosures
The following grant information was disclosed by the author:
National Research, Development and Innovation Office of Hungary: 124671.

### Competing Interests
The authors declare there are no competing interests.

### Author Contributions
- Zoltán Botta-Dukát analyzed the data, prepared figures and/or tables, authored or reviewed drafts of the paper, and approved the final draft.

### Data Availability
The code and field data are available in the Supplementary Files.

### Supplemental Information
Supplemental information for this article can be found online at http://dx.doi.org/10.7717/peerj.12763#supplemental-information.

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
