# Peer review of "Devil in the details: how can we avoid potential pitfalls of CATS regression when our data do not follow a Poisson distribution?"

_PeerJ, doi:10.7717/peerj.12763_

## Round 0.1 · original submission · Minor Revisions

Both reviewers agree that you have written a good, interesting paper with only minor weaknesses. Please consider the few comments, suggestions that both reviewers have written as these, although considered as minor, are important points that need clarification. Please also send point by point responses to these comments.

·

Basic reporting

No comment

Experimental design

There are no experiments. This is a purely theoretical paper.

Validity of the findings

No comment

Additional comments

Review of the manuscript entitled « Devil in the details : how can we avoid potential pitfalls of CATS regression when our data do not follow a Poisson distribution? » by Zoltan Botta-Dukat

This manuscript applies my CATS regression to my method of decomposing the proportional contributions of local trait-based selection and meta-community affects during community assembly. In so doing, it represents an important advance that allows for a generalization of the method to many different types of sampling distributions while still being computationally feasible, and it corrects for bias in the original formulation. Furthermore, the R code should, after some modification, make the method available to any ecologist that is familiar with R. I did not identify any major errors in logic or in the statistical details and the text is generally well written. The following points are minor corrections and suggestions that will further improve the manuscript.
First, a minor correction: It is important to clearly state that the original CATS model is based on the maximum entropy formalism of Jaynes, which is a type of Bayesian model. The derivation comes from the notion of information entropy and its maximization given linear constraints (here, constraints on trait means or other moments) and does not make any distributional assumptions. It is true that the parameter estimates are identical to those obtained by maximizing the likelihood of a generalized linear model with a Poisson distribution (or, incidentally, given a multinomial distribution). It is untrue that CATS is “equivalent” to such a GLM because the maximum entropy version of CATS does not make – or require – the additional assumptions concerning distributions. However, this is a minor point because, in any empirical study, it is possible and reasonable to justify particular distributional assumptions (Poisson, negative binomial etc.); doing this will improve parameter estimation and allow the full strength of generalized linear models to be used. I have two suggestions for improvement of this manuscript.
First, it is essential that ecologists who are not experts in statistical theory be able to apply the method. Therefore, the author should carefully explain the R code by adding lots of comment lines (#) so that users can easily follow the steps and the R code.
Second, although the simulated data sets are useful and needed, it would be good to also present the full analysis of your method using an empirical data set. For instance, (Baastrup-Spohr et al., 2015) present data on many local plant communities (quadrat samples) growing in Alvars. It would be very useful to many ecologists if you could arrange to have their data and present the full analysis on this data.

Baastrup-Spohr L, Sand-Jensen K, Nicolajsen SV, Bruun HH. 2015. From soaking wet to bone dry: Predicting plant community composition along a steep hydrological gradient. Journal of Vegetation Science, 26: 619-630.

·

Basic reporting

This paper expands the framework of Shipley 2014 and Warton 2015 to non Poisson distributions, and as such should represent a useful contribution (empirical abundance data are rarely Poisson distributed, and other types of data such as cover are definitely not, as the author points out). It would perhaps be even more useful to have a modelling framework that allows for more complex designs (eg hierarchical), as for example dispersal and environmental filters may act at different spatial scales. The author has brought together various results from the statistical literature and this in itself was interesting to read.
l.228-9: “Using the classic R2 as a measure of explained variance is suitable only in OLS regression when a normal distribution is assumed”. I don’t understand the assumption of a normal distribution – what matters is how it is defined/calculated (see Kvålseth 1985), and then interpreted depending on eg what is predicted/explained (Cox and Wermuth 1985). See Gelman et al. (2019) for some additional discussion and definition. The different definitions of R2 (as for example when you compare your equations 15 an 25) are discussed eg in Kvålseth for different models.
l. 239 “Although this procedure is correct, it […] results in slightly different values for each run”. I would not see this as a problem if the different values reflect the uncertainty of the estimate (as in eg a Bayesian R2, Gelman et al. 2019). R2 or IC are estimates and understanding the associated uncertainty is useful (it is often larger than people would believe).
l. 332: “Unfortunately, this adjustment is valid for only Gaussian distributions”. Again I do not think this is correct. If you read Fisher (1925) the argument is purely geometrical, and there is no assumption of a normal distribution as far as I could see. The adjustment (usually called after Ezekiel, not sure exactly why) has been evaluated on small samples in different papers, most recently Hittner (2020) I think.
For the whole discussion on R2, it is not impossible in the linear case to get for example that R2(trait)+R2(offset)- R2(trait; offset) is negative (that is the whole model – trait and offset – explains more than the sum of each model). This is called suppression I think (see eg Bertrand and Holder 1988) – not common in a multidimensional setting but not impossible (we encountered it in this paper: Randin et al. 2009). I don’t know if that is possible with other definitions of R2 in nonlinear models.
Details:
L212 «for a for Poisson…” remove for before Poisson

Bertrand, P. V., & Holder, R. L. (1988). A Quirk in Multiple Regression: The Whole Regression can be Greater Than the Sum of Its Parts. Journal of the Royal Statistical Society. Series D (The Statistician), 37(4/5), 371-374. doi:10.2307/2348761
Cox, D. R., & Wermuth, N. (1992). A comment on the coefficient of determination for binary responses. American Statistician, 46(1), 1-4. doi:10.2307/2684400
Gelman, A., Goodrich, B., Gabry, J., & Vehtari, A. (2019). R-squared for Bayesian Regression Models. American Statistician, 73(3), 307-309. doi:10.1080/00031305.2018.1549100
Hittner, J. B. (2020). Ezekiel’s classic estimator of the population squared multiple correlation coefficient: Monte Carlo-based extensions and refinements. The Journal of General Psychology, 147(3), 213-227. doi:10.1080/00221309.2019.1679080
Kvålseth, T. O. (1985). Cautionary Note about R2. The American Statistician, 39(4), 279-285. doi:10.2307/2683704
Randin, C. F., Jaccard, H., Vittoz, P., Yoccoz, N. G., & Guisan, A. (2009). Land use improves spatial predictions of mountain plant abundance but not presence-absence. Journal of Vegetation Science, 20(6), 996-1008. doi:10.1111/j.1654-1103.2009.01098.x

Experimental design

Not relevant here except perhaps for the 3 examples used that are described with enough details

Validity of the findings

The conclusions are clear and point to limitations of previous modelling analyses and how the proposed analyses correct for these issues.

---

## Round 0.2 · accepted · Accept

Thank you for considering and responding to all comments raised by the reviewers. I particularly thank you for the hard work you did to make the R package available and readable by a wide audience.